# Benefits of Dog-Assisted Therapy in Patients with Dementia Residing in Aged Care Centers in Spain

**DOI:** 10.3390/ijerph18041471

**Published:** 2021-02-04

**Authors:** Eva Vegue Parra, Jose Manuel Hernández Garre, Paloma Echevarría Pérez

**Affiliations:** 1Health Sciences PhD Program, Campus de los Jerónimos nº135, Universidad Católica de Murcia (UCAM), Guadalupe, 30107 Murcia, Spain; evegue@ucam.edu; 2Department of Political Sciences, Social Anthropology and Public Finance, Calle Campus Universitario, University of Murcia, s/n, El Puntal, 30100 Murcia, Spain; josemanuelhernandezgarre@gmail.com

**Keywords:** dog-assisted therapy, care centers, dementia, neurocognitive disorder, Alzheimer’s

## Abstract

(1) Background: Currently, the scientific evidence on the benefits of assisted therapy with dogs in dementia is not clear. In this study, we want to evaluate such benefits through a randomized controlled clinical trial in multiple centers across the country. (2) Methods: The participants were people over 65 years old with dementia, residing in senior centers in Spain (*n* = 334). The experimental group underwent assisted therapy with dogs based on the Comprehensive Cognitive Activation Program in Dementia, for 8 months, with weekly sessions of 45 min. Data were collected at the commencement, middle, and end of the program, to evaluate the aspects using the Mini-Examination Cognitive, the modified Bartell Index, the Cornell Scale for Depression in Dementia and the Neuropsychiatric Inventory. (3) Results: The results show significant improvements in the experimental group versus the control group in the affective (T1 = p 0.000; T2 = p 0.000) and behavioral (T1 = p 0.005; T2 = p 0.000) aspects, with the affective aspect displaying greater progress in participants with additional depressive (*p* = 0.022) or anxiety (*p* = 0.000) disorders, shorter institutionalization periods (r = −0.222, *p* = 0.004), and those undergoing complementary psychotherapy (*p* = 0.033) or alternative therapy (*p* = 0.011). (4) Conclusions: Dog therapy is effective in improving the affective and behavioral aspects of institutionalized patients with dementia.

## 1. Introduction

The World Health Organization estimates that 47.5 million people worldwide suffer from dementia; this number is expected to increase year-by-year, with the principal cause being institutionalization of the elderly [1]. The Diagnostic and Statistical Manual of Mental Disorders (DSM) describes the main psychiatric disorders, including dementia, which has been renamed a neurocognitive disorder (NCD) and is characterized by a decline in at least one of the cognitive domains (attention, executive function, memory, learning, language, visuoperceptual and visuoconstructive functions, or social cognition). From a clinical perspective, NCD may be classified as minimal, mild, moderate, or severe according to their advancement and the manifestation of more serious symptoms [2].

The progress of an NCD can be accompanied by the appearance of psychological symptoms (hallucinations, delirium, depression, apathy or anxiety) and behavioral symptoms (agitation, inappropriate behaviors, wandering or resistance to care), which cause suffering to the patient and the persons close to them [3,4]. There are various drugs indicated for dementia (cholinesterase inhibitors and memantine) and others to treat psychological and behavioral symptoms associated with dementia (neuroleptics, antidepressants, analgesics, etc.), all of which have side effects [5]. There are several studies that relate comorbidity and polypharmacy in people with dementia [6]. Therefore, potentially beneficial non-pharmacological methods are increasingly sought. This is the case of dog-assisted therapies (DAT), which are discussed here.

Human–animal interaction has been studied since the mid-20th century, and currently animals may be incorporated into therapeutic, educational, or social programs for various purposes. Thus, we understand DAT as being structured, goal-oriented therapies that are used by healthcare professionals and intentionally incorporate dogs in health with the purpose of obtaining therapeutic benefits and improving health and well-being. DAT affords opportunities for motivational, educational, and recreational benefits to improve quality of life [7].

In recent years, several studies have been conducted on the effectiveness of DAT in older people with NCD. Improvements have been observed in psychiatric symptoms such as agitation, aggressiveness, anxiety, depression, or apathy [8,9,10,11,12,13,14,15,16], improvements in social interaction with increased prosocial behavior [9,11], improved quality of life [17,18,19,20], or preservation of functions such as alertness [21] or blood pressure [22]. There are several theories that attempt to explain the observed benefits of these interventions. Some of these theories emphasize the activation of basic psychological processes such as attention, perception, or motivation in the presence of the animal and their importance for the achievement of therapeutic objectives [23].

Reviews conducted for the purpose of comparing results found much variability in the studies, making them difficult to compare. In selecting studies, many small ones are excluded based on quality, with few remaining to be examined. Some studies concluded that the improvements observed are not significant [9,13,14,15,16,19]. Studies with significant results cannot be replicated. Reasons for this may be excessively small samples (65 participants being the largest), little detail in the protocols, or variability in them (some use weekly sessions, others semiweekly, and the duration also varies) or different design (some are controlled and randomized, others not). In their conclusions, they pointed out the need for studies with a larger sample, better design, and more detail in the protocols used [24,25,26].

To that effect, the aim of the present study was to evaluate the benefits that a DAT program could provide for the affective, behavioral, cognitive, and functional aspects of patients with NCDs residing in care centers for the elderly in different provinces of Spain, through the use of a protocol based on the stimulation of basic psychological processes that is detailed and homogeneous, as well as a larger sample.

## 2. Materials and Methods

### 2.1. Design

Based on systematic reviews, we believe that the problems of replication and extrapolation of the results of current studies come from the need to obtain larger samples, better design and more detail in the protocols used [24,25,26]. We consider it especially important to improve this to be able to conclude whether DAT are beneficial for patients with NCD. For that reason, we start from the following hypothesis: DAT provide significant improvements for patients with NCD in the affective, behavioral, functional, and cognitive areas.

An experimental randomized controlled clinical trial was designed to answer the question of whether the DAT program provides significant benefits in the affective, behavioral, cognitive, and functional aspects to participating residents with NCD.

### 2.2. Participants

The study involved 18 centers distributed over the following Spanish provinces: Barcelona (2), Valencia (1), Cuenca (2), Alicante (1), Granada (2), Jaén (2), Sevilla (2), Badajoz (1), Toledo (1), Madrid (2), La Coruña (2). After excluding residents with a history of animal allergies or phobias, participation was invited from those who met the following inclusion criteria: 

- Aged over 65 years. 

- Diagnosed with an NCD and an evident cognitive deterioration, with a Mini Mental State Examination (MMSE) score of less than 25.

- Signed informed consent. 

The application of these criteria yielded a sample of 371 participants, comprising 277 women and 93 men, ultimately reduced to a total of 334 after participant losses, as shown in the flow diagram (Figure 1). The degree of NCD was taken from center-supplied data as well as the clinical histories. The sample was randomized into two groups: the experimental group, which received the DAT and the usual therapies offered by the center, and the control group, which received the center’s usual treatments but not the DAT. The participants were stratified in accordance with their degree of cognitive deterioration, gender, age, and other independent variables, in order to ensure the homogeneity of the groups.

The therapies that were carried out in the participating centers and that we classified as usual therapies are occupational therapy and physical therapy that took place every day, and to a lesser extent, psychology, socio-cultural animation, and complementary therapies that took place once or twice a week. None of the centers had had dog-assisted therapy for at least a year.

### 2.3. Procedure

The DAT was implemented progressively by the centers between September 2018 and January 2019, with each center starting as it progressed through the established steps. Each team was composed of a DAT technician and their therapy dog. The formation of the teams was homogeneous and specific to the intervention program.

The dogs were selected according to the following criteria: good health, high sociability, high impulse control, and high learning ability. They were then trained by their technicians on the specific exercises of the program (walking on a leash, handling, positions, maintenance, and skills) through a friendly methodology based on various techniques of learning by conditioning, both emotional and socially cognitive.

The training of the technicians was specific in the preparation of their dogs and in the work with elderly people with NCD, in addition to the preparation of the sessions to be carried out.

A health professional from each center was responsible for monitoring and evaluation throughout the program.

An ad hoc program was created and implemented in all participating centers, based on the recommendations of the Institute for Elderly and Social Services (IMSERSO), which advocates a comprehensive approach to address all basic psychological processes (motivation, emotion, perception, attention, language, memory, thought, and learning) [27,28]. The program consisted of weekly sessions of 45 min, conducted in groups of 10 people plus the dog and the DAT handler, over a period of 8 months. The first sessions implied the creation of a link with a gradual and positive introduction to the dog’s world. The following sessions consisted of working on the areas under study through direct interaction with the dog and a final phase of farewell. Table A1 lists the objectives of the program and some related exercises and Table A2 includes examples of sessions with different objectives.

The strategies used in the sessions according to the objectives to be worked on were the following:

- Affective area: caressing and brushing the dog on a table or chair, positive communication by the professional towards the dog and the participants, encouraging the expression of positive emotions and promoting interaction with the dog and the rest of the participants in the group.

- Behavioral area: promotion of sustained attention to the dog and to the activity the professional is conducting with the dog, varied activities within each session, proposing alternative behaviors related to the dog when unadaptive behaviors are detected, respect for personal space in each need, promotion of relaxation through petting and brushing the dog directly on the table or chair.

- Functional area: activities aimed at working the upper body, coordination and fine motor skills with the dog with a central element, through imitation, order requests, collaboration in sequences and games with the dog.

- Cognitive area: work on spatial and temporal orientation at the beginning and end of each session. Promotion of attention and concentration of the dog to facilitate its participation in the tasks. Work on reminiscence and memory stimulation related to animals from their past. Language and thought stimulation through word formation games, sayings or categorization with the dog as the central element of the activity (with puzzles, bibs, and specific toys with which the dog and the participants interact).

### 2.4. Instruments and Variables

Data collection was conducted in the beginning, at four months, and at eight months of intervention by occupational therapists or working psychologists from the participating centers. For the measurement of the four areas of study or dependent variables, the following scales were used (all widely used in the centers and studies consulted) [8,9,10,11,12,13,14,15,16,17,18,19,20,21,22]: for the cognitive area, the Mini-Cognitive State Examination (MEC-30) with a weighted kappa index of 0.637; a sensitivity of 89.8% and specificity of 75.1% [29]; for the functional area, the modified Barthel index (kappa between 0.47 and 1.00 with respect to the interobserver and between 0.84 and 0.97 for the intraobserver) [30]; for the effective area, the Cornell Scale of Depression in dementia (CSDD) (kappa between 0.61 and 0.84 and a total reliability of 0.93, internal consistency of 0.81) [31,32], and for the behavioral area, the Neuropsychiatric Disorders Inventory (NPI) with a Pearson’s correlation index of 0.879 for the severity scale and 0.92 for the stress scale [33]. In addition, the following independent sociodemographic variables were collected from the clinical records: sex, age, education (basic, middle and higher) and center, and the clinical data collected included: type of disorder additional to dementia (cardiovascular disease, depression, anxiety and diabetes), time of residence, and regular daily therapies (occupational therapy, physical therapy) and weekly complementary therapies (psychological, socio-cultural animation and alternative therapy: music therapy, work therapy and geronto-gymnastics). To avoid information bias, the double-blind technique was applied so that neither the professionals who collected the data nor the analyst knew the exposure group from the control group.

### 2.5. Ethics

The project was performed in accordance with the Helsinki Declaration and the UCAM Ethics Committee, which issued its approval (code CE031820). The staff at the centers provided verbal and written information about the study to the residents and their families and obtained informed consent.

The therapy dogs were prepared specifically for the purpose and met the national and provincial standards applying to animal welfare, health requirements, and the mental, social, and emotional well-being of the dogs [34].

### 2.6. Data Analysis

All the data were collated into an Excel database (Microsoft, Redmond, WA, USA), and analyzed using the IBM SPSS statistical software (IBM, Armond, NY, USA). It was confirmed that the observations were independent, that the samples obeyed a normal distribution, and that the variances were uniform per Kolmogorov–Smirnov analysis and Levene’s test.

The level of therapy effectiveness was measured in different centers simultaneously. The evolution of the four measurement scales was analyzed at each stage of the program using *t*-tests for related samples, and the scoring differences between the experimental and control groups were analyzed via *t*-tests for independent samples. Effectiveness was also analyzed in terms of the independent variables, with the experimental group showing progress in its scores on the significant aspects according to gender, illness, type of complementary therapy (Student’s *t*-test for independent samples), age, years of residence (Pearson’s correlation coefficient), educational level, and residential center (single-factor ANOVA and Tukey’s HSD).

## 3. Results

Table 1 sets out the frequency distribution of the different independent variables, giving average scores of 13.3 (+7) for the MMSE, 8.4 (+6.9) for the CSDD, 10.91 (+19.8) for the NPI, and 49.53 (+31.4) for the Barthel index.

Significant differences are observed in the evolution of the scores recorded for the four aspects under study, both for the experimental group and for the control group at four and eight months, but with different trends. While the affective (CSDD) and behavioral (NPI) aspects progressed positively for the experimental group and negatively for the control group, the cognitive (MMSE) and functional (Barthel) aspects show a negative progression in both groups (Table 2). The first columns correspond to the averages of the three moments (T0, T1 and T2), and the following correspond to the differences between moment 0 and 1 (T0–T1) and between moment 1 and 2 (T1–T2), to observe the progression. 

At the commencement of the program, no significant differences were observed between the experimental and control groups in the four aspects under study; however, at four and eight months, there was a significant improvement in the scores of the experimental group versus the control group in the affective (T1 = p 0.000; T2 = p 0.00) and behavioral (T1 = p 0.00 and T2 = p 0.005) aspects (Table 3).

In the post hoc analysis, we compared the significant differences found in the affective and behavioral areas with the independent variables. The sociodemographic variables that were examined, such as sex (*p* = 0.448 for the affective area, *p* = 0.393 for the behavioral area), age (*p* = 0.873 for the affective area, *p* = 0.342 for the behavioral area), and level of studies were not significant. In addition, comparisons were made between the centers without finding significant differences. The significant differences found are reflected in Table 4. We carried out two types of analysis: the *t*-test for independent samples for contrast with respect to additional pathologies and psychology and complementary therapies, and the Pearson correlation test for contrast with respect to time of residence. The cross-correlation of the two significant aspects (affective and behavioral) with the independent variables shows greater progress in the affectivity scores of patients who have an additional pathology such as depression (*p* = 0.022) or anxiety (*p* = 0.00) or who are receiving complementary psychotherapy (*p* = 0.033) or alternative therapy (*p* = 0.011). Patients with a shorter period of institutionalization also show greater progress in affectivity scores (r = −0.222, *p* = 0.004) (Table 4).

## 4. Discussion

The data show that DAT in patients with NCD is effective for both the affective and the behavioral area, significantly improving scores over a control group that follows an inverse trend of progressive worsening. This improvement shows a dose-response association, and better results are observed with longer exposure time. In the cognitive and behavioral area, however, a worsening is observed both in the exposure group and in the control group, although the deterioration is less for the exposure group during the whole intervention for the cognitive area. In the same way, it is observed that the user profile that shows the best evolution in the scores of the affective area is the patient with additional disorders of a depressive or anxious nature, who receives psychological or complementary therapy and who has been institutionalized for less time.

If we compare our study with previous research and systematic reviews, we can point out:

- In line with the recommendations of previous studies [24,25,26], the sample was increased from 65 subjects to 334 [14], and the exposure time was similar to that of the longer-term study [19].

- Our results coincide in the significant improvement in the affective and behavioral areas with several studies in which significant progression were also evident [8,10,11,12,17,18]. These studies show significant improvements in SPCD, such as agitation and aggressiveness, which belong to the behavioral area, and depression, which belongs to the affective area. In all of them, the results are significant improvements; however, other studies point to improvement trends that were not significant [9,13,14,15,16,17,19,20]. Given the results obtained in our study, we understand that the diversity in the mentioned studies may be due to the small sample analyzed, since in all of them an improvement is observed. We can say that the assisted therapy with dogs is a good complement for the improvement in the SPCD.

- As for the lack of improvement in the cognitive and functional areas, our results also coincide with the conclusions of the systematic reviews [24,25,26], although in our case, a slight deterioration is observed over time. 

In summary, it can be concluded that the best evidence in the affective and behavioral areas can have a positive influence at the psychosocial level and on the quality of life [17,18,19,20,21], which provides more data on the potential of these interventions as a complement to and support for the theories related to motivation, attention, social support, learning, and activation of the experimental system developed to date [23]. 

Although this study presents a larger sample than the previous ones, more extensive research is recommended to support the evidence. The protocol followed was based on that already studied by Gallardo Schall and Rivas Espinoza [8]. We used this protocol as a basis because one of the authors designed it and it turned out to be promising, though we adapted it to weekly sessions and the number of sessions was increased. We consider this study important, albeit with limitations: we did not consider whether the participants have had a dog before, although we know that in the sample there are participants who have had a dog before and others who have not. This may be something to consider in future research. Nevertheless, we consider that this study can be a starting point for future, more specific research since the results show, in a general way, that the protocol used is effective at an affective and behavioral level. We believe that it is useful to determine the areas sensitive to the intervention, the dose, or temporality of the intervention and the profile of the patient who benefits most from the protocol.

## 5. Conclusions

Based on the above findings, we conclude that assisted therapy with dogs has beneficial effects in people with institutionalized dementia at an affective and behavioral level. It can be useful to palliate affective and behavioral symptoms frequent in residents of senior centers if we incorporate it as a complement to the interventions already carried out in such centers.

## Figures and Tables

**Figure 1 ijerph-18-01471-f001:**
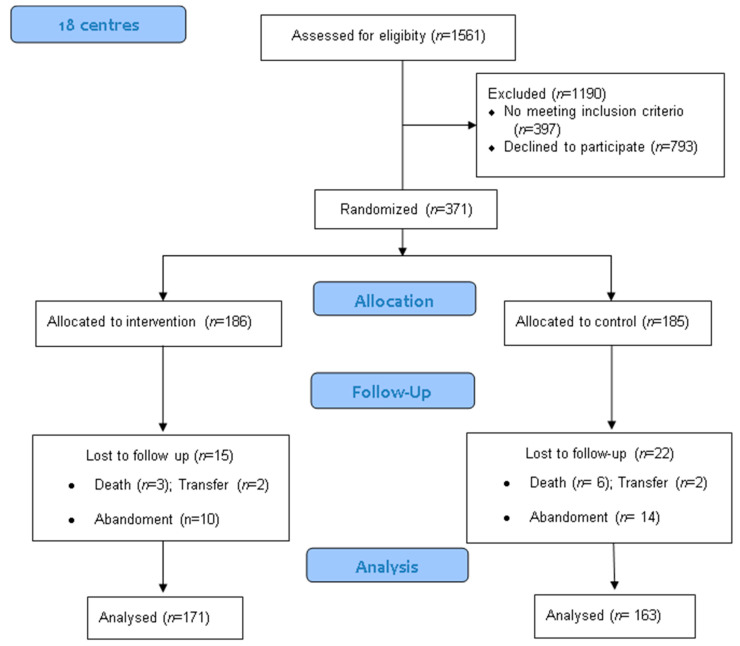
Participant flow chart.

**Table 1 ijerph-18-01471-t001:** Description of participants.

Characteristics	(*n* = 334)	%
Women	259	77.5
Men	75	22.5
Higher education	14	4.2
Intermediate education	13	3.9
Basic education	235	70.4
None	72	21.6
Alzheimer-type dementia	181	54.2
Senile-type dementia	141	42.2
Other dementias	12	3.6
Cardiovascular disease	147	44
Depression	68	20.4
Anxiety	36	10.8
Diabetes	4	1.2
OT	242	72.5
Physiotherapy	214	64.1
Psychotherapy	56	16.8
Mood improvement	91	27.2
Complementary therapies	29	8.7
Initial evaluation	$\bar{X}$	σ
MMSE	13.3	7
CSDD	8.14	6.9
NPI	10.91	19.8
Barthel	49.53	31.4

*n*: sample size %: percentage; OT: occupational therapy; $\bar{X}$ mean; σ: standard deviation, MMSE: Mini-Mental State Examination; CSDD; Cornell Scale for Depression in Dementia; NPI: Neuropsychiatric Inventory.

**Table 2 ijerph-18-01471-t002:** Evolution of groups.

	DAT	*n*	T0	T1		T0–T1		T1	T2		T1–T2	
DV			$\bar{X}$ σ	$\bar{X}$ σ	95% CI	t	*p*	$\bar{X}$ σ	$\bar{X}$ σ	95%CI	t	*p*
CSDD	Experimental	171	8.14	5.38	2.2	11.1	0	5.38	4.33	3.2	14.2	0
			6.92	5.9	3.2			5.9	5.4	4.3		
	Control	163	8.14	8.88	−1.12	−3.99	0	8.88	9.34	−1.62	−5.6	0
			6.91	7.1	0.38			7.1	7.1	−0.78		
NPI	Experimental	171	10.67	6.29	3.1	6.59	0	6.29	5.3	4	7.8	0
			19.2	15.8	5.7			15.8	14.8	6.7		
	Control	163	11.32	12.03	−1.52	−1.69	0.092	12.03	12.9	−2.7	−3	0.003
			20.5	21.1	0.12			21.1	22.2	−0.56		
MMSE	Experimental	171	13.54	13.18	−2	1.26	0.21	13.18	13.03	−0.1	1.6	0.107
			7.05	7.8	0.93			7.8	7.8	1.1		
	Control	163	13.04	12.23	0.32	3.29	0.001	12.23	11.9	0.59	4.2	0
			6.98	7.2	1.29			7.2	7.2	1.63		
Modified Barthel	Experimental	171	51.46	48.27	1.7	4.3	0	48.27	47.6	1.9	3.9	0
			31.3	31.3	4.6			31.3	31.8	5.7		
	Control	163	47.49	44.69	1.25	3.56	0	44.69	43.7	2.06	4.4	0
			31.4	32.1	4.36			32.1	32.1	5.47		

DV: dependent variable; CSDD: Cornell Scale for Depression in Dementia; NPI: Neuropsychiatric Inventory; MMSE: Mini-Mental State Exam; DAT: dog-assisted therapy; *n*: sample size; T0: initial evaluation; T1: intermediate evaluation; T2: final evaluation; $\bar{X}$: arithmetic mean σ: standard deviation; %: percentage; CI; confidence interval; t: test statistic; *p*: significance.

**Table 3 ijerph-18-01471-t003:** Comparison between groups.

T	DV	DAT	*n*		σ	95%	CI	t	*p*
				$\bar{X}$		Lower	Upper		
T0	CDSS	Experimental	171	8.14	6.92	−1.48	1.49	0.07	0.995
		Control	163	8.14	6.91				
	NPI	Experimental	171	10.67	19.15	−4.91	3.62	−0.29	0.766
		Control	163	11.32	20.49				
	MMSE	Experimental	171	13.54	7.05	−1	2.01	0.66	0.51
		Control	163	13.04	6.98				
	Barthel	Experimental	171	51.46	31.32	−2.78	10.71	1.15	0.249
		Control	163	47.49	31.36				
T1	CDSS	Experimental	171	5.38	5.97	−4.91	−2.09	−4.89	0
		Control	163	8.88	7.09				
	NPI	Experimental	171	6.29	15.83	−9.73	−1.72	−2.81	0.005
		Control	163	12.03	21.09				
	MMSE	Experimental	171	13.18	7.86	−0.67	2.58	1.15	0.25
		Control	163	12.23	7.23				
	Barthel	Experimental	171	48.27	31.32	−3.24	10.4	1.03	0.303
		Control	163	44.69	32.1				
T2	CDSS	Experimental	171	4.34	5.36	−6.39	−3.61	−7.09	0
		Control	163	9.34	7.41				
	NPI	Experimental	171	5.3	14.84	−11.71	−3.6	−3.72	0
		Control	163	12.96	22.21				
	MMSE	Experimental	171	13.03	7.78	−0.51	2.71	1.34	0.18
		Control	163	11.93	7.19				
	Barthel	Experimental	171	47.64	31.75	−2.96	10.78	1.12	0.264
		Control	163	43.73	32.11				

T: time; CSDD: Cornell Scale for Depression in Dementia; NPI: Neuropsychiatric Inventory; MMSE: Mini-Mental State Exam; DV: dependent variable; DAT: dog-assisted therapy; n: sample size; T0: initial evaluation; T1: intermediate evaluation; T2: final evaluation; $\bar{X}$: arithmetic mean σ: standard deviation; %: percentage; CI; confidence interval; t: test statistic; *p*: significance.

**Table 4 ijerph-18-01471-t004:** CDSS versus disorders, therapies, and time of residence.

		DV	CSDD			Average Progress T0–T2		
IV		*n*	T0–T2	σ	95%	CI	t	*p*
					Lower	Upper		
Depressive disorder	YES	32	−8.66	16.08	0.58	7.49	2.31	0.022
	NO	139	−4.62	6.26				
Anxiety disorder	YES	15	−14.13	21.69	4.99	14.21	4.11	0
	NO	156	−4.53	6.23				
Psychotherapy	YES	28	−3.97	3.68	−2.14	−0.09	−2.18	0.033
	NO	143	−2.89	2.15				
Complementary therapy	YES	12	−6.25	3.55	0.61	4.68	2.57	0.011
	NO	159	−3.6	3.43				
Period of residence		*n*	$\bar{X}$ CSDD	σ	$\bar{X}$ years	σ	CP	*p*
		171	4.34	5.36	1.06	1.69	−0.222	0.004

IV: independent variable; DV: dependent variable; CSDD: Cornell Scale for Depression in Dementia; *n*: sample; $\bar{X}$: arithmetic mean; T0: initial evaluation; T2: final evaluation; T0–T2: difference in scores at commencement and end of program; σ: standard deviation; %: percentage; CI: confidence interval t; test statistic; *p*: significance; YES: meets the IV criterion; NO: does not meet the IV criterion; CP: Pearson correlation.

## Data Availability

Not applicable.

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
