# Peer review of "Benefits of Dog-Assisted Therapy in Patients with Dementia Residing in Aged Care Centers in Spain"

_ijerph, 2021, doi:10.3390/ijerph18041471_

Round 1

Reviewer 1 Report

This was an interesting, clearly written and accessible paper. I think that the findings will be of interest to the sector. The authors clearly indicate that this paper is original particularly regarding the sample size. My comments are very brief in the light of this being an effective paper.

I would like to have known more about the intervention itself - what were the details of the dogs - eg age/ breed, and what was the nature of their preparation etc. What were the welfare considerations that were made?

Could the authors provide a little more information regarding the detail of the intervention - I assume handlers were present - did they interact with the participants in any way?

I'm interested in whether the authors feel that the gender balance may have had any impact on data?

Also, was it possible to find out how many participants had been dog owners in the past? Would this have had an impact on the data?

Minor point re language - Line 57 researches should be 'research'

Author Response

The corrections that have been made are as follows:

- Perhaps we have summarized too much the intervention made. In the procedure section (lines 116-128) you can see the extension we have made. As you will see, the dogs were selected by criteria of health, sociability, impulse control and learning ability; we did not consider the breed, since we consider it not relevant. Although there are breeds more used than others, more and more all kinds of breeds are used if they meet the criteria above and the preparation is adequate. We have not used age as a criterion either because we have given priority to the health of the dog, regardless of its age, although we must point out that dogs older than 9 years old have not participated because they already had some health problem. (line 121-130). We have also extended the description of the protocol carried out by means of the description of strategies in the article (line 131-161) and a complementary table with two examples of sessions. (Appendix B)

- Regarding whether gender balance has had any impact on the data: an analysis of all independent variables was conducted, but only significant results were included. We have included a paragraph indicating this (lines 247-252)

- The next question regarding whether the participants had a dog: we have pointed this out as a limitation of our study since it is a fact that we have not considered in the analysis. We can say that, within the sample, we know that there are participants who previously had a dog and others who did not. This is something to be studied in future research.

- Line 57 corrected change research by research.

Reviewer 2 Report

This study evaluated the impact of an animal-assisted therapy (AAT) program for individuals with dementia. Participants (N = 371) were enrolled from 18 care centers in Spain. The AAT program involved weekly group interactions with a therapy dog and the dog's handler over an 8-month period. Outcomes were evaluated after 4 and 8 months and compared to a control group (although no information was provided about the control condition). The authors conclude that benefits were observed in affect and behavior for participants in the AAT group. 

There certainly is value in better understanding how AATs might be used in treatment of dementia. In spite of the importance of this type of work, there are problems with this manuscript. Major concerns (organized by the section of the manuscript) are noted below.

Abstract

  • There is important information missing from the abstract. For example, it is not clear what the authors mean by "a full simulation program of the basic psychological processes" (p. 1). Similarly, results indicate differences between the experimental and control groups, but the methods did not indicate that multiple groups/conditions were part of this study. The abstract should be revised to provide a clear and consistent summary of the study.

Introduction

  • In reviewing current treatments for dementia, the authors need to provide more information on the concerns associated with medication use. As the manuscript currently stands, the authors provide no information about the concerns associated with "polymedication" (p. 1). Moreover, they do not provide any information about the most common treatments currently used and concerns or limitations related to those treatments.
  • Because this study is focused on the potential clinical benefits of animal-assisted therapies (AATs), the authors need to provide more specific information (with appropriate citations/references) about them.
  • The authors offer only general comments and do not provide any specific review of other published studies focused on AATs for dementia. This information is important for clarifying why this specific study is needed.
  • While the authors identify a goal of this research, they do not state specific hypotheses. Specific research questions/hypotheses are needed in the Introduction.

Materials and Methods

  • Figure 1 should be revised. It would be helpful if the authors used the CONSORT flow diagram as a template in revising this figure.
  • There is no description of the difference(s) between the experimental and control groups. There is a very brief description of the dog-assisted program (with additional information provided in an appendix). Detailed information on both groups (including the specific activities and timeline for those activities) must be provided.
  • More information is needed on each of the primary outcome measures used in the study. Descriptions of each measure (e.g., specific information on the constructs they assess, psychometric characteristics) are needed.
  • In terms of the data analytic plan, it appears that the repeated administration of the assessment battery (at Time 1, Time 2, and Time 3) was not incorporated into the plan and that data at each time point were evaluated as though they were independent. In addition, there are concerns when using this relatively large number of pairwise comparisons (over 20 in this study) without adjusting the p-value. Finally, the authors did not acknowledge that the data were nested within care centers.

Results

  • This section relies heavily on tables, some of which are difficult to understand. For example, the authors need to more clearly explain/describe the information presented in Table 4.

Discussion/Conclusion

  • Because of missing information related to the study's methods and results, it is challenging to know if their conclusions are appropriate/justified.
  • The authors did not offer any commentary on the limitations of their study. This information should be provided.
  • Only one brief paragraph (p. 8) was provided to contextualize this study's results. More information is needed to clarify how the results from this study fit into the broader literature.

Author Response

The corrections that have been made are as follows:

- Regarding the meaning of "a full simulation program of the basic psychological processes" there may be a translation error. The protocol is based on the recommendations of the Institute of Elderly and Social Services (IMSERSO) of Spain, which advocates for a comprehensive stimulation of people with dementia based on the activation of their basic psychological processes (motivation, perception, attention, emotion, memory, learning, language and reasoning). As we wanted to talk about the basic psychological processes in the summary, perhaps we did not make a good translation. We have corrected this, using the same terminology as the IMSERSO and detailing it in the methodology section (line 11-14).

- Regarding not commenting on the conditions in the methodology part of the summary, we consider that it is not necessary. We have a maximum of 200 words for the summary, so we have tried to choose each word. At the beginning of the summary, we say that it is a clinical, randomized and controlled study, so we understand that it is clear that there is an experimental and a control group.

- With respect to polymedication, we have included one more reference to be able to expand on this. We do not enter an explanation of this, because we did not want to extend it into something that is not the central theme of the research, but we recognize that it requires a little more explanation. We have corrected it, as well as the most common treatments, to make it better understood (lines 40-46). Our intention is to highlight the increasingly frequent use of non-pharmacological therapies to avoid or reduce such polypharmacy.

- About the background of TADs, we find it interesting to expand the information, so we have added five references and have gone deeper into the current problem. We have talked about the existing variability of the designs and results, concluding the same as the current systematic reviews and, finally, we explain the hypothesis of our study with greater justification with respect to the above. (lines 55-77)

- Figure 1 corrected using the CONSORT flowchart model.

- The description of the groups is found in Table 1, since in the analysis of the groups they were homogeneous at the beginning of the intervention, we consider that it is a table that describes both groups. The only difference between the groups is the TAD received by the experimental group. The rest of the usual therapies are received by both groups. In each center there was an experimental group and a control group, thus controlling the variations of habitual therapies that may exist between the centers. For a better understanding of this we have specified what are usual therapies (physical and occupational therapy, with daily sessions) and complementary therapies (psychology, sociocultural animation and others, with sessions of 1 or 2 a week) (Line 111-115) In addition, we have extended the description of the protocol carried out (line 117-161)

- In the instruments used we have added more information about each one, all of them are instruments widely used in the centers and in the previous studies (line 163-182).

- - In the data analysis we have incorporated some explanations of the tables (lines 196, 222-224, 249-257) and have modified Table 2 to make it more visual. As you can see, we made the t-contrasts for dependent samples between moments 0 and 1; and 1 and 2. We have also specified the Post-hoc studies that we carried out and that were not significant, we have not included them in the Table in order not to extend too much (lines 247-252).

- Considering these modifications, we hope that our conclusions will be of interest to you. We have tried to make an article with all the information we have obtained without extending ourselves too much, but if you think it is necessary, we can extend the information.

Reviewer 3 Report

Brief summary

This paper presents a randomized controlled clinical trial of a dog-assisted therapy program for people living in care homes and who suffer from dementia.

Broad comments

The strength of the study presented in this paper is the relatively large sample size (n=334) – especially for the sub-field of animal-assisted therapy. There was the appropriate control and experimental group and a set of clear experimental measures. The paper is written in a straightforward and clear way. Below I indicate a few areas that could be described in a bit more detail and some minor editing suggestions.

Specific comments 

Other studies, as referenced in the Introduction, have explored dog-assisted therapy. A few more details could be mentioned about some of the other findings and/or review papers in this section. What is meant my “mutually contradictory” (line 55)? This study has a large sample – a weakness mentioned in other studies – but did any use the same measures? Why were the 4 measures included here specifically chosen? A brief example with more details of one of the therapy sessions would be helpful (lines 101-103). The discussion could also be expanded and tie-back to some of the other findings in the DAT literature.

Editing suggestions

  1. Do not need the word “the” in the 1st sentence of the Abstract (line 9).
  2. The phrasing of sentence 1 in the Introduction makes it sound like cause of dementia is being institutionalized (I think it’s the reverse that is true) (lines 27-29).
  3. Typo on line 49.
  4. “Researches” (line 51 & 57) should be “research”.

Author Response

The corrections that have been made are as follows:

We have expanded the background so that the reader has more context to read the article. (line 55-77). Our intention is to expose the existing problems, the studies carried out are small, remarkably diverse in terms of the protocol used and the research designs, so the systematic reviews agree on the need for more research and higher quality. With this as a starting point, our study aims to answer the most basic question: what benefits TAD brings to patients with dementia, for which we consider a large sample and a controlled and randomized experimental design to be necessary.

- "Mutually contradictory" (line 55) is an error in the translation of the article. We wanted to say that existing studies obtain non-replicable results, since studies on the same benefits obtain different results. Such small samples, different protocols and designs may be the cause of this. We have corrected this by explaining it in more precise words.

- We decided to measure the effects in the four areas because they encompass all the benefits studied so far. We want to see the fundamental basis of these interventions and, from there, go deeper into more concrete objectives.

- (Lines 101-103) We have expanded this information with the general strategies carried out with the dog and the complementary table (Appendix B) with two examples of sessions, to provide more detail of the protocol since we consider it to be one of the fundamental things.

- Line 9 "the" corrected and changed the first sentence by a sentence that introduces the problem of the diversity of existing studies, we have removed the word institutionalization by residents or care centers that is more concrete. (lines 9-12)

- Lines 27-29: it is a typographical error, change "the" by "of

- Lines 55 and 57: typographical error corrected.